# Mid- and Long-Term Surgical Outcomes Due to Infective Endocarditis in Elderly Patients: A Retrospective Cohort Study

**DOI:** 10.3390/jcm11226693

**Published:** 2022-11-11

**Authors:** Jill Jussli-Melchers, Mohamed Ahmed Salem, Jan Schoettler, Christine Friedrich, Katharina Huenges, Gunnar Elke, Thomas Puehler, Jochen Cremer, Assad Haneya

**Affiliations:** 1Department of Cardiovascular Surgery, University Hospital of Schleswig-Holstein, Campus Kiel, Arnold-Heller-Str. 3, Hs C, D-24105 Kiel, Germany; 2Department of Anesthesiology and Intensive Care Medicine, University Medical Center Schleswig-Holstein, Campus Kiel, D-24105 Kiel, Germany

**Keywords:** infective endocarditis, cardiac surgery, cardiopulmonary bypass, elderly patients

## Abstract

Background: Infective endocarditis (IE) is one of the true remaining dreaded situations in cardiovascular medicine. Current international guidelines do not include specific recommendations for treatment options of infective endocarditis (conventional vs. surgical) based on the patient’s age, functional status or comorbidities. Elderly patients have less invasive and often delayed surgeries compared to younger patients due to their shorter long-term survival probabilities. In the setting of IE, this might not be the right treatment, as surgery is the only curative option in up to 50% of all endocarditis patients. The aim of our study was to evaluate the mid- and long-term surgical outcomes due to infective endocarditis of patients aged ≥70 years. Methods: Between 2002 and 2020, a retrospective study with 137 patients aged 70 years and older and 276 patients aged below 70 years was conducted. Altogether, 413 consecutive patients who received surgery due to infective native or prosthetic valve endocarditis were assigned to either the elderly (E)-Group or the control (C)-Group. Primary endpoints were short- and long-term MACCEs (Major Adverse Cardiac and Cerebrovascular Events) as a composite of death or major adverse events, and secondary endpoints were intraoperative variables and postoperative course. Results: Preoperative risk factors differed significantly. Elderly patients had more arterial hypertension, atrial fibrillation, diabetes, chronic renal insufficiency and coronary heart disease. Fewer of them were in a state of emergency. Time from diagnosis to OR, antibiotic pretreatment, length of surgery and cardiopulmonary bypass time were significantly longer in the E-Group. Furthermore, 44.5% of patients in the E-Group had prosthesis endocarditis as opposed to 29.7% in the C-group. During postoperative follow-up, new onset of hemodialysis, duration of ventilation, delirium, reintubation and tracheotomy rates were significantly higher in the E-Group. There were significant differences in 7- and 30-day mortality. One- year survival was 62% for the E-Group and 79% for the C-Group. Five-year survival was 47% for the E-Group and 67% for the C-Group. Conclusions: This study demonstrates that surgery for infective endocarditis is a high-risk procedure, especially for elderly people. Nevertheless, as it is more or less the only concept to increase long-term survival, it should be offered generously to all patients who are still able to take care of themselves.

## 1. Introduction

The incidence of infective endocarditis (IE) in the general population is continually increasing [1]. Epidemiological studies in Europe suggest that one of the main reasons for this marked rise is the steady increase in elderly patients [2,3], in addition to the increasing number of healthcare procedures [4] and use of intracardiac electronic devices or implanted valve prostheses [1]. Up to 25% of IE cases are deemed to be healthcare-acquired [5]. Surgical treatment in the general endocarditis population is required in about 25–30% of acute cases and in another 20–40% of subacute and chronic cases [6]. Patients younger than 65 years receive surgical treatment in 46% of cases—i.e., the expected need—while 29% of patients between 65 and 79 years proceed to surgery, and only 5.8% of patients older than 80 years do so [1]. Thus, it is most likely that the indication for surgery is undertriaged in patients older than 70 years of age [1].

The aim of our study was to evaluate (1) the short- and long-term rate of MACCEs (Major Adverse Cardiac and Cerebrovascular Events) as a composite of death or major adverse events (myocardial infarction, stroke or repeat valve procedures) and (2) intraoperative variables (extracorporeal circulation time, cross-clamp time and number of valves involved) and postoperative course (ventilation time, drainage loss and acute renal failure) in patients aged ≥70 years compared to patients <70 years.

## 2. Materials and Methods

### 2.1. Patients and Study Design

In this single-center, retrospective cohort study, patients aged ≥70 years with IE located in at least one valve or valve prosthesis who proceeded to surgery between 2002 and 2020 were included. Patients with conservative, non-surgical treatment were excluded. Definition of active endocarditis was ongoing antibiotic therapy. Patients aged younger than 70 years served as a control (C) group with the same eligibility criteria defined for the elderly (E) group.

Data were collected and extracted from the institution’s database and from medical records. The Institutional Ethics Committee of the Christian-Albrechts University Kiel approved the study protocol and authorized its conduct and follow-up (file number D 458/20). Individual written informed consent from each patient for study participation was obtained.

### 2.2. Patient Management

In our endocarditis team, which consists of a cardiologist, a cardiac surgeon and a consultant for infectious diseases, all patients were discussed. All patients of the E-Group had microbiological proof of endocarditis. In the C-Group, 274 out of 276 patients had microbiological proof. This was usually obtained by blood culture to identify the organisms according to species and sensitivities. All patients had undergone a transthoracic or transesophageal echocardiogram, in which the location and the size of vegetation, presence of valve destruction or abscess as well as left ventricular ejection fraction were analyzed. Apart from that, the diagnosis was confirmed according to the modified Duke Criteria. Antibiotic treatment started as soon as IE was plausible. If the patient’s condition was stable, a coronary angiography and additional computed tomography (CT) including cerebral CT, thoracic CT and whole-body CT scans were performed, especially in redo patients or in high-risk patients. After confirmation of diagnosis and discussion in our endocarditis team, patients were referred to our department and scheduled for near-term surgery.

An intravenous treatment regime was maintained for 4–6 weeks postoperatively if the diagnosis was intraoperatively reaffirmed. All patients with neurological complications had an evaluation of neurological status by a consultant neurologist and a computer tomography scan of the brain to estimate risks of bleeding and prognoses if they were intubated. Perioperative risk factors and intraoperative data as well as predictors for mortality were analyzed and evaluated.

### 2.3. Surgical Management

All patients were administered routine general anesthesia. Usually, median sternotomy was performed. A few patients with mitral valve endocarditis had minimally invasive anterolateral thoracotomy. All patients received curative surgery by senior surgeons. After opening the pericardium, a heart–lung machine was installed for extracorporeal circulation. This was usually conducted by arterial cannulation of the aorta and a single venous cannulation of the right atrial appendage if the aortic valve or only small parts of the mitral valve were affected, or by double cannulation of the superior and inferior vena cava if the tricuspid valve or mitral valve were operated on. This was followed by cross-clamping of the ascending aorta. Mostly, mild hypothermia (34 °C) was used. Myocardial protection was obtained by antegrade and retrograde application of cold blood cardioplegic solution. Since 2015, extracorporeal cytokine adsorption using a CytoSorb^®^ filter has been part of the routine as septic shock prophylaxis and installed in the heart–lung machine circuit. Choice of prosthesis (biological or mechanical) was, whenever possible, left to the patient’s preference. Nevertheless, the surgical method depended on intraoperative findings and the macroscopic degree of valve destructions as well as the clinical judgement of the surgical team based on conventional guidelines. According to the individual patient’s situation and the cardiac findings, additional surgical steps (myocardial revascularization, aortic replacements, PFO or VSD closure) were performed.

### 2.4. Statistical Analysis

Statistical analysis was conducted using the SPSS Statistics software (Version 24.0, Chicago, IL, USA). The normality of continuous variables was assessed using the Lilliefors test/Kolmogorov–Smirnov test. Values of continuous data are presented as mean ± standard deviation or as median with range or interquartile range when appropriate. Categorical variables are displayed as frequency distributions (n) and simple percentages (%). A univariate comparison between the groups for categorical variables was made using the χ^2^ test and Fisher’s exact test when appropriate. Quantitative variables were compared by the *t*-test or Mann–Whitney U test. The probability of event-free survival was determined on the basis of survival curves using the Kaplan–Meier method and compared using the log-rank test. Statistical significance was considered when *p* < 0.05. Variables associated with 30-day mortality were included into a multivariable logistic regression analysis for all patients (model 1), patients <70 years old (model 2) and patients ≥70 years old (model 3). The predictive value of the multivariable model was estimated using the Hosmer–Lemeshow χ^2^ test. Included variables were female gender, EuroSCORE II, LV-function < 30%, IDDM, preoperative dialysis, NYHA IV, previous cardiac surgery, cardiogenic shock, neurological deficit and presence of an abscess in model 1; for models 2 and 3, all included variables are presented in (b) in Table 1.

## 3. Results

Out of our clinical database, 413 patients were retrieved. Patients 70 years and older were allocated to the elderly group (E-Group). They had a median age of 76 years (73–79 y). Patients aged younger than 70 years had a median age of 57 years (47–64 y) and were assigned to the control group. The age difference was highly significantly different. More people were in the younger group (n = 276) than in the study group. Female gender was not significantly higher in the elderly group.

### 3.1. Patients’ Baseline Characteristics and Clinical Presentation

Patients aged 70 years and older had a more severe clinical status, which can be explained by significant differences between the two groups concerning EuroSCORE II. Male gender accounted for 70.8% of the study group and 76.4% of the control group. This was not significant. All risk factors depending on age and risk factors for coronary heart disease such as atrial fibrillation, diabetes mellitus type 2 and smoking were significantly higher in the study group, as was previous PCI. The sickness burden concerning clinical presentation and preoperative state did not differ significantly. More patients of the C-Group came as an emergency (*p* = 0.25). More of them had a fever or a liver disease, whereas more patients of the E-Group had a malignant tumor. Time from diagnosis to surgery differed significantly (*p* = 0.003), as did time from antibiotic start to surgery (*p* = 0.003), which was in both cases longer for the E-Group. The bacterial spectrum was more frequently Staph. aureus in the C-Group and enterococcus in the E-Group. Methicillin-resistant Staph. aureus (MRSA) was not significantly higher in any group. In the elderly group there was no drug abuse, whereas in the C-Group, drug abuse was 8.3% (*p* = 0.001). Affected valves were mainly the aortic valve and the mitral valve as well as prothesis endocarditis. The difference concerning affected valves was significant (*p* = 0.027). Prosthesis endocarditis occurred mainly on the mitral valve (MV) (83.3% in C-Group vs. 62.5% in E-Group), and post-TAVI endocarditis only appeared in the study group; both were not significant. There was a tendency towards a higher incidence of abscesses in the elderly group (*p* = 0.053). An overview of the demographic and clinical presentation data is outlined in Table 2.

### 3.2. Operative Data and Secondary Endpoints

The length of surgery differed significantly between the groups (*p* = 0.001), as did the cardiopulmonary bypass time (*p* = 0.04). The elderly patients received on average more red blood cell units (*p* < 0.001) and a higher number of platelet units (*p* = 0.045). The aortic valve was more often replaced by a biological prothesis in the elderly group (53.7% vs. 42.5%, *p* = 0.008) and by a mechanical prothesis in the C-Group (9.1% vs. 1.5%, *p* = 0.008). Aortic root replacement was more often performed with a biological prothesis in both groups, and no mechanical aortic root replacement was carried out in the elderly group. For the mitral valve, biological protheses were used in both groups more often. Mechanical valves were hardly used in the elderly group. Of the C-Group, 6.5% had a mitral valve reconstruction compared to 9.6% of the E-Group. Few patients had a tricuspid valve replacement. Of the C-Group, ten patients (3.6%) had a tricuspid valve reconstruction compared to two patients (1.5%) of the E-Group. No patient of our study cohort had pulmonary valve endocarditis. Fifty percent of the study group had concomitant procedures such as aortocoronary bypass surgery and ventricular or atrial septal defect closures. In both groups, twelve patients (2.9%) needed a pacemaker procedure postoperatively. Further information concerning intraoperative data and secondary endpoints can be seen in Table 3.

### 3.3. Postoperative Data and Primary Endpoints

Postoperative data are summarized in Table 4. Differences in early and late postoperative complications were noticeable between both groups.

The re-exploration rates due to profuse postoperative bleeding or cardiac tamponade (C-Group 10.8% vs. E-Group 15.6%, *p* = 0.17) did not differ significantly, but the 24-h drainage losses (500 mL [300–950] vs. 775 mL [500–1350]; *p* < 0.001) and the 24 h and 48 h numbers of fresh frozen plasma units as well as platelet units showed significant differences.

ICU stay itself (C-Group 2 [1–6] vs. E-Group 4 [2–9], *p* = 0.001) and other factors determining ICU stay such as new onset of hemodialysis (C-Group 10.6% vs. E-Group 25.6%; *p* =< 0.001), ventilation time (C-Group 14 h [8–37] vs. E-Group 23 h [11–85]), rate of reintubation (C-Group 9.7% vs. E-Group 17.4%, *p* = 0.027), rate of tracheotomy (C-Group 11.8% vs. E-Group 20.2%, *p* = 0.027), rate of bronchopulmonary infection (C-Group 7.7% vs. E-Group 17.9%, *p* = 0.002) and postoperative delirium (C-Group 10.2% vs. E-Group 28.2%, *p* < 0.001) were significantly different as well. More elderly patients (17% vs. 11.4%) developed sepsis without significance. One patient (0.9%) had a sternal wound infection in the elderly group compared to eight patients (3.2%) within the younger group.

There were highly significant differences concerning 7- (*p* = 0.04) and 30-day mortality (*p* = 0.002). There were significant differences for one-year, three-year and five-year survival as graphically demonstrated by the Kaplan–Meier curves (see Table 5 (*p* =< 0.001)).

## 4. Discussion

Larger country-wide registries found out that the rate of surgical treatment decreases dramatically with increasing age [1]. In our retrospective single-center study in which only surgically treated patients were included, we intended to find out how the outcomes of elderly patients compared to those of younger patients and which risk factors have to be balanced for a carefully considered decision for every single patient.

### 4.1. Patients’ Baseline Characteristics and Clinical Presentation

Age is known to be a risk factor concerning the outcome of nearly every therapy. Saran et al. demonstrated that age over 70 years is an exceptionally high risk factor, along with ejection fraction below 40%, body mass index over 30 kg/m^2^, chronic lung disease and diabetes [7]. We decided to stratify the data by ages below 70 years and ages of 70 years and older, as the logistic regression analysis did not show a significant impact of age quartile groups on 30-day mortality, but the cut-off of 70 years of age did.

In order to specifically evaluate age as a risk factor, no differences in gender, body mass index, ejection fraction or chronic obstructive pulmonary disease could be observed between the groups. Additive and logistic EuroSCORE I and EuroSCORE II differed significantly through age alone. In order to assess the risk factors for both age groups separately, we conducted two additional multivariable analyses. With these two additional analyses, we have quite surely eradicated a possible age gap bias.

We considered matching the data but decided against it because in clinical practice, older patients do not have the same risk factors and morbidities as younger patients. We therefore had to accept that some clinical characteristics differed widely between the groups. Nevertheless, all demographic aspects corresponded to the usual picture of elderly patients in a clinical setting. In particular, the predominance of men in both groups was already known from other studies [8,9]. We intended to depict the clinical situation as exactly as possible. Moreover, we must admit that matching would have led to a strong reduction in case numbers. We did assume that all patients were treated with comparable surgical concepts according to common practice.

In the elderly, prosthesis endocarditis is increasingly frequent, as is infestation of the mitral valve. This phenomenon has also been observed by other groups [10].

We therefore included “previous cardiac surgery” into our multivariable analysis to adjust the data for this risk factor, but we excluded the variable prosthesis IE due to insignificance (*p* = 0.613) in our first analysis. In a second analysis, we included prothesis endocarditis. In this analysis, we found female gender, dialysis, NYHA IV, cardiogenic shock, neurological deficits (TIA or stroke) and abscess as highly significant risk factors for 30-day mortality. Age of 70 years and older did not show significance in this second analysis. This also reinforces the importance of proceeding to surgery even in elderly patients.

We did not observe any cases of injection drug users with IE in the elderly group, whereas 8.3% of cases in the younger group were users. The fact that drug abuse is rather common in younger groups has been mentioned before [8]. The rate we found might be underestimated. The profile of microorganisms showed a similar distribution to other groups: Staph. aureus followed by S. viridans in the younger group and Enterococcus followed by Staph. aureus in the elderly group [10].

### 4.2. Operative Data and Secondary Endpoints

The length of surgery was significantly higher in the E-Group. This might be due to more intraoperative bleeding and delayed hemostasis. Supporting this hypothesis is the significantly higher number of packed red blood cells intraoperatively and the high amount of fresh frozen and plasma units postoperatively. The cardiopulmonary bypass time was also significantly elevated, which might be due to a higher rate of pulmonary hypertension and prolonged reperfusion time. The intraoperative choice of prothesis would rather lower the cross-clamp and bypass time.

### 4.3. Postoperative Data and Primary Endpoints

The postoperative course and outcomes differed significantly between the two groups. The elderly group more often had new onsets of hemodialysis. Ventilation time, reintubation and tracheotomy were also significantly higher in the E-Group. Others suggested that chronic dialysis and decreased left ventricular ejection fraction and pneumonia are causes for failure to wean after cardiopulmonary bypass [11]. The rate of chronic renal insufficiency before surgery was significantly higher in the E-Group. Therefore, a higher rate of new onset of hemodialysis was expectable. This might contribute to higher rates of ventilation time and reintubation, although the LVEF did not differ significantly between the groups. The incidence of pulmonary complications after cardiopulmonary bypass can be as high as 20–35% [12]. For elderly patients, we found a reintubation rate of 17.4%. More than twenty percent (20.2%) received tracheotomy. This is therefore comparatively low. Nevertheless, especially hypoxemia and acute respiratory distress syndrome come along with poor prognosis and high mortality [13,14]. The E-Group had a rate of postoperative delirium of 28.2% vs. 10.2% in the younger patients. This is comparable to other studies, in which rates of postoperative delirium between 17 and 61% have been reported. They also state age as a predisposing factor for postoperative delirium [15].

All these factors together, i.e., hemodialysis, bronchopulmonary infection, reintubation and postoperative delirium, which complicate the postoperative course are known to induce increased mortality. Consequently, the 7-day as well as 30-day and hospital mortality rates were all significantly higher in the study group than in the younger group.

Considering the survival curves of the elderly and younger endocarditis patients, there was a significant difference between the groups. One-year survival was 79% for the younger group and 62% for the elderly group. These are comparably good survival rates [16]. Untreated patients have one-year mortality rates approaching 40% [17]. Five-year survival was 67% for the control group, which has been found by other authors [18], and 47% for the elderly group. For this age group, reasonable five-year survival rates for infective endocarditis are very hard to find. However, the advantage of higher survival remained stable in both groups over the years.

## 5. Study Limitations

Our results should be interpreted with caution and viewed as hypotheses generated in light of the retrospective study design and the relatively small sample size from a single center. While treatment was performed according to guideline recommendations, it was still based on the clinical judgment of the referring physicians and of the surgical team at our center. Additionally, age lost its significance when the logistic regression analysis was performed for prothesis endocarditis.

## 6. Conclusions

Our study shows that surgical treatment for infective endocarditis in elderly patients as well as younger patients is feasible and associated with improved outcomes. Nevertheless, infective endocarditis remains a severe condition where the timing from diagnosis to antibiotic treatment and consecutive surgery leaves room for improvement in the elderly group.

## Figures and Tables

**Table 1 jcm-11-06693-t001:** (a) Logistic regression analysis for 30-day mortality in all endocarditis patients. (b) Logistic regression analysis for 30-day mortality in endocarditis patients aged < 70 years and ≥70 years.

**(a)**			
**Predictors**	**Odds Ratio**	**95% CI**	***p*-Value**
Female gender	2.115	1.119–4.001	0.021
Age ≥ 70 years	1.925	1.046–3.543	0.035
Dialysis (acute and chronic)	2.853	1.295–6.283	0.009
NYHA 4	3.065	1.553–6.049	0.001
Previous cardiac surgery	2.232	1.126–4.423	0.021
Cardiogenic shock	4.167	1.256–13.829	0.020
Neurological deficits (TIA or stroke)	3.053	1.521–6.130	0.002
Abscess	2.252	1.199–4.228	0.012
**(b)**			
**Predictors**	**Odds Ratio**	**95% CI**	***p*-Value**
Age < 70 years			
Female gender	3.280	1.355–7.937	0.008
Body mass index	1.088	1.025–1.156	0.006
Pulmonary hypertension (sPAP > 25 mm Hg)	3.901	1.518–10.023	0.005
Dialysis (acute and chronic)	3.008	1.026–8.820	0.045
Previous cardiac surgery	4.032	1.734–9.374	0.001
Cardiogenic shock	20.763	5.107–84.417	<0.001
Culture negative endocarditis	3.369	1.422–7.985	0.006
Age ≥ 70 years			
Arterial hypertension	0.196	0.058–0.663	0.009
NYHA 4	5.609	1.791–17.570	0.003
Insulin dependent diabetes mellitus	3.817	1.017–14.321	0.047
Dialysis (acute and chronic)	4.646	1.144–18.859	0.032
Cerebral embolization	5.724	1.512–21.669	0.010
Aortic valve endocarditis	0.229	0.060–0.869	0.030
Abscess	4.329	1.565–11.975	0.050

(a) CI, confidence interval; TIA, transient ischemic attack; NYHA, New York Heart Association. (b) CI, confidence interval; NYHA, New York Heart Association.

**Table 2 jcm-11-06693-t002:** Patients’ baseline characteristics.

	All Patients (n = 413)	C-Group (n = 276, 67%)	E-Group (n = 137, 33%)	*p*-Value
Age, years	61.1 ± 14.9 64 (52; 73)	53.7 ± 12.7 57 (47; 64)	75.9 ± 3.8 76 (73; 79)	<0.001
Female gender	105 (25.4%)	65 (23.6%)	40 (29.2%)	0.215
Body mass index [kg/m^2^]	25.9 (23.0; 29.4)	25.5 (22.6; 29.3)	26.2 (23.4; 29.9)	0.151
EuroSCORE II	12.1 (5.2; 27.3)	8.3 (3.6; 19.7)	24.5 (12.0; 45.9)	<0.001
COPD	50 (12.1%)	30 (10.9%)	20 (14.6%)	0.274
Arterial hypertension	240 (58.1%)	134 (48.6%)	106 (77.4%)	<0.001
Pulmonary hypertension	86 (20.9%)	49 (17.8%)	37 (27.2%)	0.026
LVEF (%),	55 (49;55)	55 (50;55)	55 (47;55)	0.381
LVEF < 30%	41 (10.5%)	23 (8.9%)	18 (13.6%)	0.147
**Heart rhythm**				
Atrial fibrillation	81 (19.6%)	40 (14.5%)	41 (29.9%)	<0.001
Pacemaker patient	40 (9.7%)	23 (8.3%)	17 (12.4%)	0.187
Peripheral vascular disease	36 (8.7%)	22 (8.0%)	14 (10.2%)	0.446
Drug abuse	23 (5.6%)	23 (8.3%)	0 (0.0%)	0.001
**Type 2 diabetes mellitus**	83 (20.1%)	34 (12.3%)	49 (35.8%)	<0.001
IDDM	45 (10.9%)	20 (7.2%)	25 (18.2%)	0.001
Hyperlipoproteinemia	116 (28.1%)	56 (20.3%)	60 (43.8%)	<0.001
Smoking	103 (27.8%)	86 (34.3%)	17 (14.2%)	<0.001
**Immunosuppressive therapy**	11 (2.7%)	10 (3.6%)	1 (0.7%)	0.110
Acute renal insufficiency	53 (12.8%)	29 (10.5%)	24 (17.5%)	0.045
Chronic renal insufficiency	116 (28.1%)	60 (21.7%)	56 (40.9%)	<0.001
NYHA IV	83 (20.2%)	53 (19.3%)	30 (22.1%)	0.519
**Coronary heart disease**	178 (43.2%)	94 (34.1%)	84 (61.8%)	<0.001
Previous PCI	37 (9.0%)	15 (5.4%)	22 (16.1%)	<0.001
**Previous cardiac surgery**	171 (41.4%)	94 (34.1%)	77 (56.2%)	<0.001
CABG	9 (2.2%)	4 (1.4%)	5 (3.6%)	
Aortic valve replacement	69 (16.7%)	33 (12.0%)	36 (26.3%)	
Mitral valve replacement/repair	6 (1.5%)	3 (1.1%)	3 (2.2%)	
Combined valve surgery	79 (19.1%)	48 (17.4%)	31 (22.6%)	
TAVI	2 (0.5%)	0 (0%)	2 (1.5%)	
Others	6 (1.5%)	6 (2.2%)	0 (0%)	
**Clinical presentation**				
Acute myocardial infarction (≤48 h)	14 (3.4%)	12 (4.4%)	2 (1.5%)	0.156
Cardiogenic shock	21 (5.1%)	14 (5.1%)	7 (5.1%)	0.987
CPR (≤48 h)	9 (2.2%)	5 (1.8%)	4 (2.9%)	0.487
**Preoperative state**				
Emergency	90 (21.8%)	69 (25.0%)	21 (15.3%)	0.025
Transfer from intensive care unit	109 (26.5%)	79 (28.7%)	30 (21.9%)	0.139
Intubated at admission	38 (9.2%)	25 (9.1%)	13 (9.5%)	0.887
**Neurological deficits**	81 (19.6%)	58 (21.0%)	23 (16.8%)	0.308
**Embolization**	114 (27.6%)	83 (30.1%)	31 (22.6%)	0.111
**Fever**	270 (66.5%)	192 (70.6%)	78 (58.2%)	0.013
1 = up to surgery	63 (15.5%)	51 (18.8%)	12 (9.0%)	
2 = until 72 h before surgery	15 (3.7%)	12 (4.4%)	3 (2.2%)	
3 = 4–7 days before surgery	39 (9.6%)	28 (10.3%)	11 (8.2%)	
4 = over 7 days	153 (37.7%)	101 (37.1%)	52 (38.8%)	
Tumor/malignancy	55 (13.3%)	29 (10.5%)	26 (19.0%)	0.017
Rheumatic disease	23 (5.6%)	17 (6.2%)	6 (4.4%)	0.458
Previous endocarditis	60 (14.5%)	43 (15.6%)	17 (12.4%)	0.389
**Liver disease**	55 (13.3%)	44 (16.0%)	11 (8.0%)	0.025
**Time from diagnosis to surgery**				0.003
1 ≤ 1 day	65 (15.9%)	53 (19.4%)	12 (8.8%)	
2 = 2–3 days	46 (11.2%)	35 (12.8%)	11 (8.0%)	
3 = 4–7 days	56 (13.7%)	40 (14.7%)	16 (11.7%)	
4 ≥7 days	243 (59.3%)	145 (53.1%)	98 (71.5%)	
**Time from antibiotic start to surgery**				0.003
1 =< 1 day	59 (14.5%)	48 (17.6%)	11 (8.1%)	
2 = 2–3 days	38 (9.3%)	28 (10.3%)	10 (7.4%)	
3 = 4–7 days	47 (11.5%)	37 (13.6%)	10 (7.4%)	
4 => 7 days	264 (64.7%)	160 (58.6%)	104 (77.0%)	
**Pathogens**				
1 = staph. aureus	82 (20.0%)	64 (23.4%)	18 (13.1%)	
2 = enterococcus	61 (14.8%)	28 (10.2%)	33 (24.1%)	
3 = streptok. viridans	43 (10.5%)	34 (12.4%)	9 (6.6%)	
4 = grampos. Streptococcus	37 (9.0%)	22 (8.0%)	15 (10.9%)	
5 = HACEK group	1 (0.2%)	1 (0.4%)	0 (0%)	
6 = mycosis	6 (1.5%)	5 (1.8%)	1 (0.7%)	
7 = other	39 (9.5%)	27 (9.9%)	12 (8.8%)	
8 = non-pathogen	113 (27.5%)	75 (27.4%)	38 (27.7%)	
9 = Staphylococcus epidermidis	28 (6.8%)	17 (6.2%)	11 (8.0%)	
10 = 2 + 7	1 (0.2%)	1 (0.4%)	0 (0%)	
MRSA	14 (3.4%)	11 (4.0%)	3 (2.2%)	0.403
**Affected valves**				0.027
1 = AV	128 (31.0%)	95 (34.4%)	33 (24.1%)	
2 = MV	92 (22.3%)	61 (22.1%)	31 (22.6%)	
3 = TV	7 (1.7%)	6 (2.2%)	1 (0.7%)	
5 = AV + MV	33 (8.0%)	24 (8.7%)	9 (6.6%)	
6 = MV + TV	2 (0.5%)	2 (0.7%)	0 (0%)	
7 = only prosthetic valve endocarditis	143 (34.6%)	82 (29.7%)	61 (44.5%)	
8 = TAVI	1 (0.2%)	0 (0%)	1 (0.7%)	
9 = AV + TV	5 (1.2%)	5 (1.8%)	0 (0%)	
10 = AV + TV + MV	2 (0.5%)	1 (0.4%)	1 (0.7%)	
**Insufficiency (at least grade II, medium) and localization**	359 (87.3%)	244 (89.1%)	115 (83.9%)	0.142
1 = AV	108 (26.3%)	89 (32.5%)	19 (13.9%)	
2 = MV	78 (19.0%)	55 (20.1%)	23 (16.8%)	
3 = TV	8 (1.9%)	6 (2.2%)	2 (1.5%)	
5 = AV + MV	32 (7.8%)	19 (6.9%)	13 (9.5%)	
6 = MV + TV	3 (0.7%)	3 (1.1%)	0 (0%)	
7 = Prostheses	71 (17.3%)	41 (15.0%)	30 (21.9%)	
8 = paravalv. leakage	17 (4.1%)	11 (4.0%)	6 (4.4%)	
9 = prosthetic endocarditis + parav. leak	17 (4.1%)	6 (2.2%)	11 (8.0%)	
10 = prosthetic endocarditis + flap	13 (3.2%)	10 (3.6%)	3 (2.2%)	
11 = parav. leak + valve	2 (0.5%)	0 (0.0%)	2 (1.5%)	
12 = more than 2 valves	10 (2.4%)	4 (1.5%)	6 (4.4%)	

**Table 3 jcm-11-06693-t003:** Intraoperative data and secondary endpoints.

	All Patients (n = 413)	Patients Aged < 70 Years (n = 276, 67%)	Patients ≥ 70 Years (n = 137, 33%)	*p*-Value
Length of surgery [min]	273 (220;355)	265 (210;338)	305 (242;385)	0.001
Cardiopulmonary bypass time [min]	166 (125;215)	161 (120;208)	176 (135;225)	0.044
Cross-clamp time [min]	116 (86;156)	111 (83;156)	122 (92;157)	0.089
Circulatory arrest [min]	0 (0–36)	0 (0–31)	0 (0–36)	0.294
Number of packed red blood cells, unit	3 (0–27)	2 (0–27)	4 (0–17)	<0.001
Number of fresh frozen plasma, unit	0 (0–13)	0 (0–12)	0 (0–13)	0.744
Number of platelets, unit	1 (0–6)	1 (0–6)	1 (0–4)	0.045
Abscess	113 (27.8%)	67 (24.7%)	46 (33.8%)	0.053
**Vegetation**	285 (70.4%)	194 (72.1%)	91 (66.9%)	0.278
1 =< 5 mm	49 (12.1%)	39 (14.5%)	10 (7.4%)	
2 = 5–10 mm	63 (15.6%)	38 (14.1%)	25 (18.4%)	
3 = 11–20 mm	134 (33.1%)	89 (33.1%)	45 (33.1%)	
4 => 20 mm	39 (9.6%)	28 (10.4%)	11 (8.1%)	
**AVR**	305 (74.2%)	201 (73.1%)	104 (76.5%)	0.461
1 = AVR biological	190 (46.2%)	117 (42.5%)	73 (53.7%)	
2 = AVR mechanical	27 (6.6%)	25 (9.1%)	2 (1.5%)	
3 = AVr	3 (0.7%)	2 (0.7%)	1 (0.7%)	
4 = Aortic root replacement biological	80 (19.5%)	52 (18.9%)	28 (20.6%)	
5 = Aortic root replacement mechanical	5 (1.2%)	5 (1.8%)	0 (0%)	
**MVR**	155 (37.7%)	102 (37.1%)	53 (39.0%)	0.711
1 = MVR biological	111 (27.0%)	73 (26.5%)	38 (27.9%)	
2 = MVR mechanical	13 (3.2%)	11 (4.0%)	2 (1.5%)	
3 = MVr	31 (7.5%)	18 (6.5%)	13 (9.6%)	
**TVR**	15 (3.6%)	13 (4.7%)	2 (1.5%)	0.159
1 = TVR biological	3 (0.7%)	3 (1.1%)	0 (0%)	
3 = TVr	12 (2.9%)	10 (3.6%)	2 (1.5%)	
**With:**	193 (47.0%)	125 (45.5%)	68 (50.0%)	0.385
1 = ACB	49 (11.9%)	32 (11.6%)	17 (12.5%)	
2 = VSD closure	1 (0.2%)	1 (0.4%)	0 (0%)	
3 = PM	12 (2.9%)	9 (3.3%)	3 (2.2%)	
4 = other	101 (24.6%)	67 (24.4%)	34 (25.0%)	
5 = ASD closure	4 (1.0%)	2 (0.7%)	2 (1.5%)	
6 = several	26 (6.3%)	14 (5.1%)	12 (8.8%)	

AVR, aortic valve replacement; AVr, aortic valve repair; MVR, mitral valve replacement; MVr, mitral valve repair; TVR, tricuspid valve replacement; TVr, tricuspid valve repair; ACB, aortocoronary bypass; VSD, ventricular septal defect; PM, pacemaker procedure; ASD, atrial septal defect.

**Table 4 jcm-11-06693-t004:** Postoperative data and primary endpoints.

	All Patients (n = 413)	Patients Aged < 70 Years (n = 276, 67%)	Patients ≥ 70 Years (n = 137, 33%)	*p*-Value
24 h drainage loss [mL]	600 (300;1100)	500 (300;950)	775 (500;1350)	<0.001
Re-thoracotomy due to bleeding/tamponade	50 (12.4%)	29 (10.8%)	21 (15.6%)	0.169
24 h number of packed red blood cells, unit	2 (0–27)	2 (0–27)	2 (0–20)	0.611
24 h number of fresh frozen plasma, unit	0 (0–29)	0 (0–29)	3 (0–18)	0.006
24 h number of platelets, unit	0 (0–8)	0 (0–8)	0 (0–5)	0.032
48 h number of packed red blood cells, unit	2 (0–27)	2 (0–27)	2 (0–23)	0.142
48 h number of fresh frozen plasma	0 (0–35)	0 (0–35)	3 (0–24)	0.006
48 h number of platelets, unit	0 (0–9)	0 (0–9)	0 (0–8)	0.004
Ventilation time [h]	16 (9;45)	14 (8;37)	23 (11;85)	<0.001
Reintubation	49 (12.3%)	26 (9.7%)	23 (17.4%)	0.027
Tracheotomy	57 (14.5%)	31 (11.8%)	26 (20.2%)	0.027
Bronchopulmonary infection	45 (11.1%)	21 (7.7%)	24 (17.9%)	0.002
New onset of hemodialysis	61 (15.6%)	28 (10.6%)	33 (25.6%)	<0.001
Hemodialysis, days	5 (3;9)	4 (3;8)	5 (3;10)	0.602
ICU time [d]	3 (1;7)	2 (1;6)	4 (2;9)	0.001
Re-admission to the ICU	34 (8.5%)	24 (9.0%)	10 (7.6%)	0.634
Re-admission POD	9.5 (5.0;16.3)	10.0 (5.0;17.0)	8.0 (4.0;14.0)	0.513
Postoperative days	10 (7;16)	10 (7;16)	10 (6;16)	0.578
Postoperative delirium	64 (16.1%)	27 (10.2%)	37 (28.2%)	<0.001
**Neurologic damage**	27 (6.8%)	14 (5.3%)	13 (9.9%)	0.085
TIA	9 (2.3%)	5 (1.9%)	4 (3.1%)	
Stroke	18 (4.5%)	9 (3.4%)	9 (6.9%)	
CPR	22 (5.5%)	15 (5.6%)	7 (5.3%)	0.903
Newly appeared atrial fibrillation	17 (4.9%)	7 (2.9%)	10 (9.4%)	0.010
Pacemaker patient	47 (11.6%)	27 (10.0%)	20 (14.9%)	0.142
Postoperative myocardial infarction	5 (1.3%)	3 (1.1%)	2 (1.5%)	0.666
Sepsis	54 (13.3%)	31 (11.4%)	23 (17.0%)	0.114
Sternal wound infection	9 (2.5%)	8 (3.2%)	1 (0.9%)	0.282
7-day mortality	50 (12.1%)	27 (9.8%)	23 (16.8%)	0.040
30-day mortality	74 (17.9%)	38 (13.8%)	36 (26.3%)	0.002
Hospital mortality	68 (16.6%)	36 (13.1%)	32 (23.5%)	0.008
Cardiac death	10 (14.3%)	5 (13.5%)	5 (15.2%)	
Cerebral death	1 (1.4%)	1 (2.7%)	0 (0%)	
Sepsis	9 (12.9%)	5 (13.5%)	4 (12.1%)	
Multi-organ failure	50 (71.4%)	26 (70.3%)	24 (72.7%)	

**Table 5 jcm-11-06693-t005:** Survival curves of elderly and younger endocarditis patients.

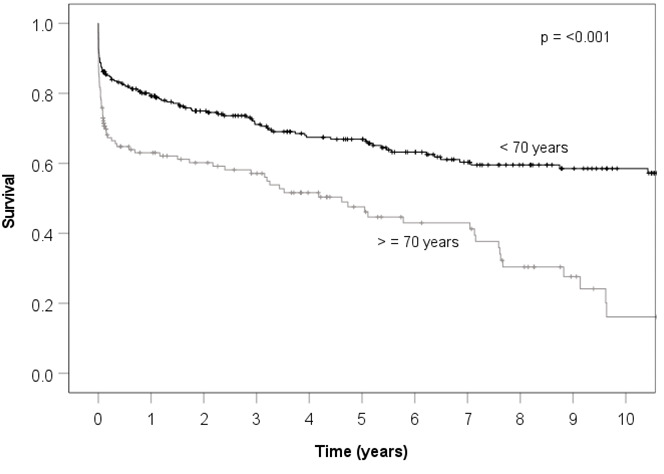					
Time (years)	1	3	5	10	
Patients < 70 years					
N at risk	263	158	121	50	6
Survival	0.79	0.71	0.67	0.58	0.53
Patients ≥ 70 years					
N at risk	128	59	39	8	
Survival	0.62	0.56	0.47	0.17	

## Data Availability

Data are available upon requirement.

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
