# Peer review of "Mid- and Long-Term Surgical Outcomes Due to Infective Endocarditis in Elderly Patients: A Retrospective Cohort Study"

_jcm, 2022, doi:10.3390/jcm11226693_

Round 1

Reviewer 1 Report (Previous Reviewer 2)

Dear Authors

Thank you for your thorough and comprehensive review, Although I'm still not absolutely convinced as to whether discrimination of the results by < and > 70 years is the est approach to include age as a relevant influence in diagnostics and therapy your data provide many interesting aspects which are worth being published.

Reviewer 2 Report (Previous Reviewer 1)

Authors did their best and I accept their effort for revision. 

This manuscript is acceptable for publication. 

This manuscript is a resubmission of an earlier submission. The following is a list of the peer review reports and author responses from that submission.

Round 1

Reviewer 1 Report

Dear authors

This manuscript is about the surgical outcomes due to infective endocarditis focusing elderly patients. Recent TAVR era inevitably is strongly associated with infective endocarditis(IE) from extension of usage from younger to very old population, so we have to pay a close attention to IE including classic prosthesis implantation. Authors wrote manuscript very well, but I have some questions and comments.

1. In page 3, line 130-131, authors described as followings: ‘They had a median age of 76 years (73- 79y). Patients aged younger than 70 years had a median age of 57 years (47-64y) and were assigned to the control group. Between E and C group, because the gap of median age is too big, late 70`s is very different status compared with late 50`s (about 20 years gap), and actually you have to match by IPTW or propensity score matching method. This age gap can make uncorrectable bias such as mortality, morbidity, so I recommend authors to classify enrolled population by quartile groups according to age for sub-analysis (ex. <55, 55~65, 65~75, 75<)

2. Considering patients’ characteristics focusing prosthesis endocarditis, there is very big different between groups as followings: Previous cardiac surgery – Total 171 (41.4%) C-group 94 (34.1%) and E group 77 (56.2%), p <0.001. Also for only prosthetic valve, there were total 143 (34.6%) endocarditis and of them, 82 (29.7%) were C- group and 61 (44.5%) were E- group. In real operation situation, redo operation from prosthesis IE might be a more risk and higher hazard cases actually, so this differences could be a serious statistical bias for survival or hazard complication comparison study. So authors have to make a comments for adequate statistical comparison method in discussion.

3. In terms of 30-day mortality, there were total 74 (17.9%) cases in hospital mortality cases and of them, 38 (13.8%) were C- group and 36 (26.3%) were E-group. In comparison with reported mortality from other studies, relatively high mortality in E-group and main cause of death was multi-organ failure (24,72.7%). The delay of surgical management from diagnosis to surgery decision is very important recently and early surgery is recommended from many highly impacted studies even cerebral embolic event cases. So I recommended authors have to clearly describe institutional strategy for surgical management for IE regardless patients` age.

Author Response

Dear reviewer:

Thank you for your valuable comments on our article. We addressed each of your single critique points clarifying your issues.

COMMENT 1: In page 3, line 130-131, authors described as followings: ‘They had a median age of 76 years (73- 79y). Patients aged younger than 70 years had a median age of 57 years (47-64y) and were assigned to the control group. Between E and C group, because the gap of median age is too big, late 70`s is very different status compared with late 50`s (about 20 years gap), and actually you have to match by IPTW or propensity score matching method. This age gap can make uncorrectable bias such as mortality, morbidity, so I recommend authors to classify enrolled population by quartile groups according to age for sub-analysis (ex. <55, 55~65, 65~75, 75<)

ANSWER to comment 1: We thank the reviewer for this important comment. We had discussed these topics when we designed this study. Since logistic regression analyses did not show a significant impact of age quartile groups on 30-day mortality, but a cutoff of 70 years did, we decided to stratify the data by age < 70 years and age ≥ 70 years.

We considered to match the data, but decided against it, because in clinical practice older patients do not have the same risk factors and morbidities as younger patients. We intended to depict the clinical situation as exact as possible. Moreover, we must admit that matching would have led to a strong reduction in case numbers.

To assess the risk factors of both age groups separately, we have conducted two additional multivariable analyses. In the main with these two additional analyses we are quite sure to have eradicated the age gap bias. We therefore think that our study keeps its relevance.

CHANGES: In “2.4 Statistical Analysis” we have added “Variables associated with 30-day mortality were included into multivariable logistic regression analysis for all patients (model 1), patients < 70 years (model 2) and patients ≥ 70 years (model 3)” see line 83 and 84 and line 89 and 90”.

Over and above that another table was added showing the Logistic regression analysis for 30-day mortality in endocarditits patients aged < and ≥ 70 years.

Table 5b: Logistic regression analysis for 30-day mortality in endocarditis patients aged < and ≥ 70 years

Predictors

Odds ratio

95% CI

p-value

Age < 70 years

Female gender

3.280

1.355-7.937

0.008

Body mass index

1.088

1.025-1.156

0.006

Pulmonary hypertension (sPAP > 25 mm Hg)

3.901

1.518-10.023

0.005

Dialysis (acute and chronic)

3.008

1.026-8.820

0.045

Previous cardiac surgery

4.032

1.734-9.374

0.001

Cardiogenic shock

20.763

5.107-84.417

<0.001

Culture negative endocarditis

3.369

1.422-7.985

0.006

Age ≥ 70 years

Arterial hypertension

0.196

0.058-0.663

0.009

NYHA 4

5.609

1.791-17.570

0.003

Insulin dependent diabetes mellitus

3.817

1.017-14.321

0.047

Dialysis (acute and chronic)

4.646

1.144-18.859

0.032

Cerebral embolization

5.724

1.512-21.669

0.010

Aortic valve endocarditis

0.229

0.060-0.869

0.030

Abszess

4.329

1.565-11.975

0.050

-------------------------------------------------------------------------------------------------------------------------------

COMMENT 2: Considering patients’ characteristics focusing prosthesis endocarditis, there is very big different between groups as followings: Previous cardiac surgery – Total 171 (41.4%) C-group 94 (34.1%) and E group 77 (56.2%), p <0.001. Also, for only prosthetic valve, there were total 143 (34.6%) endocarditis and of them, 82 (29.7%) were C- group and 61 (44.5%) were E- group. In real operation situation, redo operation from prosthesis IE might be a more risk and higher hazard cases actually, so these differences could be a serious statistical bias for survival or hazard complication comparison study. So, authors have to make a comments for adequate statistical comparison method in discussion.

ANSWER to comment 2: We also thank you for this valuable comment. We had already included “Previous cardiac surgery” into our multivariable analysis to adjust the data for this risk factor, but we excluded the variable prosthesis IE due to insignificance (p=0.613) in the former analysis. We have now conducted a new analysis, also including prosthesis IE. Age > 70 years fails significance in this new analysis.

Alternative Table 5a including prothesis endocarditis: Logistic regression analysis for 30-day mortality in endocarditis patients

Predictors

Odds ratio

95% CI

p-value

Female gender

2.034

1.075-3.848

0.029

Age ≥ 70 years

1.753

0.904-3.400

0.097

Dialysis (acute and chronic)

2.441

1.031-5.780

0.042

NYHA 4

2.892

1.443-5.793

0.003

Previous cardiac surgery

3.039

0.995-9.287

0.051

Cardiogenic shock

6.230

1.829-21.218

0.003

Neurological deficits (TIA or stroke)

3.126

1.548-6.313

0.001

Abscess

2.243

1.185-4.248

0.013

First step: age ≥ 70 Jahre, female gender, LVEF poor (< 30), IDDM (Typ I und Typ II

on insulin), dialysis preoperative acute and chronic (0=no, 1=yes), previous cardiac surgery

(0=no, 1=yes), cardiogenic shock (0=no, 1=yes), neurological deficits (0=no, 1=yes),

abscess (0=no, 1=yes), only prothesis E7, Euroscore II [%]

 CHANGES: We outlined the different analyses in the Discussion and added it to the Limitations section.

-------------------------------------------------------------------------------------------------------------------------------

COMMENT 3: In terms of 30-day mortality, there were total 74 (17.9%) cases in hospital mortality cases and of them, 38 (13.8%) were C- group and 36 (26.3%) were E-group. In comparison with reported mortality from other studies, relatively high mortality in E-group and main cause of death was multi-organ failure (24,72.7%). The delay of surgical management from diagnosis to surgery decision is very important recently and early surgery is recommended from many highly impacted studies even cerebral embolic event cases. So, I recommended authors have to clearly describe institutional strategy for surgical management for IE regardless patients` age.

ANSWER to comment 3: We also thank you for this valuable comment. You are right that delay of surgery may lead to higher 30-day mortalities and that time from diagnosis to surgery is very important. The aim of our study was to show, that elderly people can be treated in the same way as younger patients and that they may benefit from early surgery. In the introduction we already assumed, that “surgery is undertriaged in patients 70 years and older” (Please see line 11 and 12). We also had the feeling that infective endocarditis in elderly people is underdiagnosed or that diagnosis is delayed which we could prove in this study.

We are a centre in a German “Bundesland” with few cardiothoracic surgeries. Proceeding to surgery is usually not referrers first idea. Therefore, patients often reach our centre with some delay. Also, diagnosis of infective endocarditis is often more difficult to find in elderly patients than in younger patients.

Usually, it is proceeded to surgery as soon as an infective endocarditis patient reaches our department.

Our findings concerning time from diagnosis to surgery and time from antibiotic treatment to surgery (“Time from diagnosis to surgery differed significantly (p=0.003) as did time from antibiotic start to surgery (p=0.003), which was in both cases longer for E-Group see line 111/112”) let us to the conviction that surgery due to infective endocarditis might be beneficial for all patients regardless of age.

CHANGES: We added line 46-48 in 2.2 Patient Management to clarify that patients were treated as soon as they were referred to our department. (“After confirmation of diagnosis and discussion in our Endocarditis team patients were referred to our department and scheduled for near-term surgery”.)

Reviewer 2 Report

Dear Authors,

I have read your manuscript with the greatest interest. It is obvious that endocarditis is one of the major problems in cardiac surgery and it is steadily growing. Therefore, it is very commendable to perform a more in-depth analysis of the overall profile of the patients as well as the specific risk factors. Although I appreciate your work, I have some concerns:  I believe that the age distinction is somewhat arbitrarily chosen. As can be seen from your logistic regression analysis (Table 5), age 70 years or older is not the strongest predictor of early mortality. Rather, it is the weakest of all. Age is clearly a risk factor, but it would be much more useful to analyze the entire patient population together, focusing on the risk factors for early and late mortality. You can of course choose a relevant discriminating factor afterwards, but then it would be better to do propensity score matching where the factor of interest is the only difference. I suggest that you recalculate the statistics with the entire cohort and do a comprehensive risk factor analysis. This would then include type of bacteria, procedural characteristics, etc. 

Author Response

Dear reviewer:

Thank you for your major and minor criticism concerning our manuscript. We included additional information to improve our manuscript hoping to address all your concerns.

COMMENT 1: I have read your manuscript with the greatest interest. It is obvious that endocarditis is one of the major problems in cardiac surgery and it is steadily growing. Therefore, it is very commendable to perform a more in-depth analysis of the overall profile of the patients as well as the specific risk factors. Although I appreciate your work, I have some concerns:  I believe that the age distinction is somewhat arbitrarily chosen. As can be seen from your logistic regression analysis (Table 5), age 70 years or older is not the strongest predictor of early mortality. Rather, it is the weakest of all. Age is clearly a risk factor, but it would be much more useful to analyze the entire patient population together, focusing on the risk factors for early and late mortality. You can of course choose a relevant discriminating factor afterwards, but then it would be better to do propensity score matching where the factor of interest is the only difference. I suggest that you recalculate the statistics with the entire cohort and do a comprehensive risk factor analysis. This would then include type of bacteria, procedural characteristics, etc. 

ANSWER to comment 1: Thank you very for your concerns regarding our data stratification by age below 70 years and age ≥ 70 years. We discussed this item excessively beforehand and performed a logistic regression analysis, in which no significant impact of age quartile groups on 30-day mortality could be proven, but a cutoff of 70 years did. Therefore, we decided to stratify the data by age < 70 years and age ≥ 70 years.

To assess the risk factors of both age groups separately, we have conducted two additional multivariable analyses. In the main with these two additional analyses we are quite sure to have eradicated the age gap bias. We therefore think that our study keeps its relevance.

We considered to match the data, but decided against it, because in clinical practice older patients do not have the same risk factors and morbidities as younger patients. We intended to depict the clinical situation as exact as possible. Moreover, we must admit that matching would have led to a strong reduction in case numbers.

Risk factor stratification has also already been done in our group.

Please be so kind to read: Friedrich C, Salem M, Puehler T, et al. Sex-Specific Risk Factors for Short- and Long-Term Outcomes after Surgery in Patients with Infective Endocarditis. J Clin Med. 2022;11(7):1875. Published 2022 Mar 28. doi:10.3390/jcm11071875.

In this analysis age was an important factor for 30-day mortality. Therefore, we decided to have a closer look on age especially, as we had the strong feeling that the number of elderly patients in our department is continuously increasing.

CHANGES: In “2.4 Statistical Analysis” we have added “Variables associated with 30-day mortality were included into multivariable logistic regression analysis for all patients (model 1), patients < 70 years (model 2) and patients ≥ 70 years (model 3)” see line 83 and 84 and line 89 and 90”.

Over and above that another table was added showing the Logistic regression analysis for 30-day mortality in endocarditits patients aged < and ≥ 70 years.

Table 5b: Logistic regression analysis for 30-day mortality in endocarditis patients aged < and ≥ 70 years

Predictors

Odds ratio

95% CI

p-value

Age < 70 years

Female gender

3.280

1.355-7.937

0.008

Body mass index

1.088

1.025-1.156

0.006

Pulmonary hypertension (sPAP > 25 mm Hg)

3.901

1.518-10.023

0.005

Dialysis (acute and chronic)

3.008

1.026-8.820

0.045

Previous cardiac surgery

4.032

1.734-9.374

0.001

Cardiogenic shock

20.763

5.107-84.417

<0.001

Culture negative endocarditis

3.369

1.422-7.985

0.006

Age ≥ 70 years

Arterial hypertension

0.196

0.058-0.663

0.009

NYHA 4

5.609

1.791-17.570

0.003

Insulin dependent diabetes mellitus

3.817

1.017-14.321

0.047

Dialysis (acute and chronic)

4.646

1.144-18.859

0.032

Cerebral embolization

5.724

1.512-21.669

0.010

Aortic valve endocarditis

0.229

0.060-0.869

0.030

Abszess

4.329

1.565-11.975

0.050
